# Geospatial Analysis of COVID-19: A Scoping Review

**DOI:** 10.3390/ijerph18052336

**Published:** 2021-02-27

**Authors:** Munazza Fatima, Kara J. O’Keefe, Wenjia Wei, Sana Arshad, Oliver Gruebner

**Affiliations:** 1Department of Geography, The Islamia University of Bahawalpur, Punjab 63100, Pakistan; munazza.fatima@iub.edu.pk (M.F.); sana.arshad@iub.edu.pk (S.A.); 2Department of Epidemiology, Epidemiology, Biostatistics, and Prevention Institute, University of Zurich, CH-8001 Zürich, Switzerland; kara.okeefe@gmail.com (K.J.O.); wenjia.wei@uzh.ch (W.W.); 3Department of Geography, University of Zurich, CH-8057 Zürich, Switzerland

**Keywords:** spatial analysis, COVID-19, disease mapping, exposure mapping, spatial epidemiology, health geography

## Abstract

The outbreak of SARS-CoV-2 in Wuhan, China in late December 2019 became the harbinger of the COVID-19 pandemic. During the pandemic, geospatial techniques, such as modeling and mapping, have helped in disease pattern detection. Here we provide a synthesis of the techniques and associated findings in relation to COVID-19 and its geographic, environmental, and socio-demographic characteristics, following the Preferred Reporting Items for Systematic reviews and Meta-Analyses extension for Scoping Reviews (PRISMA-ScR) methodology for scoping reviews. We searched PubMed for relevant articles and discussed the results separately for three categories: disease mapping, exposure mapping, and spatial epidemiological modeling. The majority of studies were ecological in nature and primarily carried out in China, Brazil, and the USA. The most common spatial methods used were clustering, hotspot analysis, space-time scan statistic, and regression modeling. Researchers used a wide range of spatial and statistical software to apply spatial analysis for the purpose of disease mapping, exposure mapping, and epidemiological modeling. Factors limiting the use of these spatial techniques were the unavailability and bias of COVID-19 data—along with scarcity of fine-scaled demographic, environmental, and socio-economic data—which restrained most of the researchers from exploring causal relationships of potential influencing factors of COVID-19. Our review identified geospatial analysis in COVID-19 research and highlighted current trends and research gaps. Since most of the studies found centered on Asia and the Americas, there is a need for more comparable spatial studies using geographically fine-scaled data in other areas of the world.

## 1. Introduction

The epidemic intensity of COVID-19 has been strongly shaped by crowding, evidenced by higher prevalence in big cities as compared to smaller cities and rural areas [1], but the pandemic’s immense spread has also been enabled by other biological, social, environmental, and economic factors. Various epidemiological studies have been carried out to explore the spatial spread of COVID-19. Every country has been affected differently and has exhibited a unique pattern of incidence and mortality under the influence of many underlying factors. Temporal and spatial variation in the incidence rate of COVID-19 have produced the three distinctive high-rate clusters, initially China, then Western Europe, and finally the USA [2,3].

Study of disease distribution and diffusion over time and space is a core theme of both health geography and spatial epidemiology [4]. Spatial analysis is essential to understanding the spatial spread of infection and its association with the community and environment, especially in the early stages of an outbreak [5]. The spatial and spatial-temporal proximity concept is deeply linked with the transmission of infectious diseases, since transmission rates are more likely to be high when people are near each other [6].

Spatial patterns of disease often suggest links between disease and potential risk factors of a geographic area [7], and the analysis of such patterns can be traced back to the cholera epidemic in London in 1854, when John Snow mapped cholera-related deaths, which resulted in subsequent removal of handles from public water pumps on Broad Street [8]. 

Technically, spatial analytical methods can be classified into three categories: disease mapping, exposure mapping, and spatial epidemiological modeling [9]. Disease mapping, or visualization, describes geographic patterns of diseases. Exposure mapping explores the spatial covariance of potential influencing factors driving disease outcomes. Spatial epidemiological modeling aims to estimate or predict health outcomes while adjusting for potential influencing factors and the spatial structure of the data.

There is massive research available about COVID-19, changing and improving our understanding on a daily basis. A previous study on the major themes of geospatial techniques [10] took an interdisciplinary perspective on COVID-19 and reviewed relevant scholarly work published through the end of May 2020. Our current study will expand on the previous review and will further synthesize spatial studies of COVID-19. Our main objectives will be to: (1) review the use of spatial analysis tools and techniques in investigations of the geographic variation of the COVID-19 pandemic and its potential influencing factors (environmental, socio-economic, demographic, and healthcare-related); and (2) synthesize study results separately for three categories: disease mapping, exposure mapping, and spatial epidemiological modeling.

## 2. Methods

For this scoping review, we performed a MEDLINE search via the PubMed database, applying the Preferred Reporting Items for Systematic reviews and Meta-Analyses extension for Scoping Reviews (PRISMA-ScR) [11]. We used the following MeSH (Medical Subject Headings) terms: ‘Spatial Analysis’ OR ‘Geographic Mapping’ OR ‘Spatial Regression’ OR ‘Space time Clustering’ OR ‘Spatio Temporal Analysis’ AND ‘COVID-19′. The search was carried out on 30 September 2020. We included published, peer-reviewed journal articles based on the spatial analysis of COVID-19. Studies were considered eligible for inclusion if they broadly described the use of spatial analysis techniques for studying and analyzing the COVID-19 pandemic. We included only English-language articles published between 1 January 2020 and 30 September 2020. Manuscripts describing both qualitative and quantitative study types were considered. Clinical trials were not considered for this review.

Based on our search strategy, 74 articles were identified, and a two-stage screening was then carried out. At the first stage of screening, only the title and abstract were evaluated as to whether they fulfilled the eligibility criteria. At the second stage, the selected articles were screened and reviewed completely in terms of main spatial analysis techniques used, geographic extent of the study, mapping software, and COVID-19 data used in the study and relationships with other determinants (reported cases, confirmed cases, tested population, demographic aspects of patients, etc.). 

Keyword frequency in the included articles was then analyzed and a word cloud diagram was constructed via an online service (https://www.wordclouds.com) (Accessed on 1 November 2020). Subsequently, we categorized the findings according to spatial epidemiological terminology, that is, disease mapping, exposure mapping, and spatial epidemiological modeling.

## 3. Results

In total, 74 articles matched the search criteria (Figure 1). Of these, 34 were excluded at the first stage and two at the second stage. For the 38 remaining articles, the most frequent months of publication were June (n = 12) and August (n = 12), followed by May (n = 8). We noted fewer article publications in March, April, and September (n = 2, n = 3, and n = 1, respectively). None of the articles was published in January or February.

Figure 2 displays a word cloud of keywords found in the reviewed articles. “COVID-19” was the most frequently used word, along with its other parallel names like “SARS-CoV-2”, “coronavirus”, “pandemic”, “disease”, or “health.” The terms “GIS”, “spatial”, “analysis”, “mapping”, “vulnerability”, “social”, “socioeconomic”, and “infectious” also appeared frequently as keywords used in articles. We noted that most of the studies were of “ecological” nature and that “geographic”, “spatio-temporal”, “space–time”, “surveillance”, and “cluster” analysis were repeatedly used as keywords for indicating spatial techniques used in the included articles. Asia and the Americas were the geographic regions where the most included studies were carried out.

Table 1 shows details for study foci, spatial techniques, software used, and geographic extent of the selected articles, and Table 2 shows the main findings of the selected articles. ArcGIS (Environmental Systems Research Institute: Redlands, CA, USA) [12] was the most frequently used software for spatial statistical analysis of data (n = 21), followed by the statistical software and environment R(R Foundation for Statistical Computing: Vienna, Austria) [13] (n = 9), GeoDa (The Center for Spatial Data Science at the University of Chicago, Chicago, IL, USA) [14] (n = 6), Quantum GIS (Open Source Geospatial Foundation: Beaverton, OR, USA) [15] (n = 7), SaTScan (Kulldorff M. and Information Management Services Inc: Calverton, MD, USA) [16] (n = 5), and TerraView (Image Processing Division, Brazilian National Institute of Space Research: Cuiabá, Brazil) [17] (n = 3). We noted that most of the studies used a combination of two or more spatial statistical software products. Some articles declared their statistical analysis software as SPSS (IBM: Armonk, NY, USA) [18] (n = 5) and MS Excel (Microsoft Corporation: Redmond, WA, USA) [19] (n = 2). Epidemic Location Intelligence System (http://epidemija.gis.ba/webcity/#/) (Accessed on 26 February 2021) (n = 1) and Mapbox^2^ (n = 1) were also mentioned.

China was the most frequent geographic location of the studies (n = 12), followed by Brazil (n = 9), USA (n = 8), and Italy (n = 2). One study from each of Australia, Bosnia and Herzegovina, Portugal, South Korea, Spain, Kenya, and Germany was also identified. However, not a single study was found from South Asia or the Middle East.

### 3.1. Disease Mapping

We observed that disease mapping approaches ranged from unadjusted to adjusted disease outcomes in space and time, with studies controlling for sex, age, occupation, and healthcare services. For example, during the early months of the epidemic, Huang, Liu and Ding [26] used a simple COVID-19 heat map approach to show the spatial patterns of the epidemic in China. Lioa et al. [30] studied COVID-19 in disadvantaged areas of Liangshan Prefecture, China. Ponjavić et al. [34] applied geospatial visualization techniques to study the distribution of COVID-19 incidence rates in Bosnia and Herzegovina on a series of maps using ELIS (Epidemic Location Intelligence System). In another study by Michelozzi et al. [32], spatial distribution of COVID-19 incidence was shown on a map, while the rest of the analysis was largely based on statistical analysis.

Spatial autocorrelation techniques were also used by many researchers. For example, hotspot analysis was used by Tang et al. [37] to show the spatial intensity of infection in China. Similarly, another study carried out in China quantified the spatial distribution of COVID-19 by hotspot analysis [23]. Likewise, Li et al. [30] analyzed clusters of COVID-19 incidence in China by using global and local Moran’s I in the ArcGIS environment. In the Hubei province of China, spatiotemporal analysis of COVID-19 daily cases and their incidence rate was done by Yang et al. [38], using hotspot and cluster analysis in ArcGIS. A simple ecological study by Cavalcante and Abreu [22] explored spatial variation in COVID-19 incidence and mortality rates in Rio de Janeiro, Brazil, using global and local Moran’s I. The software packages used for mapping were QGIS and GeoDa.

Many researchers combined geographic visualization of COVID-19 distribution with additional influencing factors as well, including age, sex, occupation, transport, mobility pattern, and environmental factors. For example, Gao et al. [24] mapped the spatial distribution of COVID-19-infected healthcare workers in China using QGIS. Likewise, another descriptive study focused on occupational health and showed the spatial distribution of total COVID-19 cases in relation to orthopedic surgeons of patients over 60 years of age in the USA [27].

Researchers also applied smoothing techniques. For example, a study by Pedrosa and Albuquerque [33] found that the spatial distribution of intensive-care bed capacity was significantly associated with COVID-19 in the State of Ceará, Brazil by using smoothed Bayesian estimators. Rex et al. [35], in their exploratory study, analyzed the spread of COVID-19 in the state of São Paulo, Brazil by using kernel density estimation (KDE) and related the spread of the virus to population mobility, mainly through roads and air transport.

R-based statistical modeling of temperature and COVID-19 cases in Spain found no significant associations [21]. Another study showed the spatial distribution of population outflow from Wuhan to the rest of China, along with evidence for an association between population outflow and COVID-19 cases [28]. Geo-temporal progression of the epidemic was highly structured in China, a study by Rivas et al. [36] found. Their study revealed that network properties, including synchronicity and directionality, determined the epidemic profiles observed in several Chinese provinces, fostering the planning and implementation of more precise and locally specific interventions.

Space–time clustering techniques were also used by researchers to identify propagation and prediction of COVID-19. For example, Hohl et al. [25], aiming to facilitate surveillance and improve resource allocation and decision-making, found weekly clusters of COVID-19 in the USA by using the Poisson space–time scan statistic implemented in SaTScan and presented their findings in an interactive web application developed in R Shiny. Andrade et al. [20] used a prospective space–time scan statistic to detect spatiotemporal clusters of transmission of COVID-19 in Sergipe, Brazil with QGIS, TerraView, and SaTScan. In addition, using global Moran’s I and space–time scan statistic, Kim and Castro [29] provided evidence that the South Korean government’s epidemic response measures were significantly associated with changes in COVID-19 clusters.

### 3.2. Exposure Mapping

A study led by Macharia et al. [42], using ArcGIS zonal statistics and arithmetic means, found that COVID-19 risk was heterogeneously distributed across multiple social epidemiological indicators in Kenya—including elderly populations, malnutrition, smoking, living in informal settlements—and comorbidity of obesity, hypertension, or diabetes. Lakhani [41] used hotspot (Getis-Ord Gi*) analysis in ArcGIS and showed that disability and access to health services were risk factors for COVID-19 in an elderly population in Melbourne, Australia. A study in Rio De Janeiro, Brazil by Santos et al. [44] used ArcGIS to determine that city neighborhoods with higher average household density, high tuberculosis incidence, and large older populations (>60 years) were more vulnerable to COVID-19 infections. Gomes et al. [40] used a combination of various spatial statistical techniques in a set of diverse software packages (Joint Point Regression Program for time trend analysis, SaTScan, TerraView, and QGIS) to investigate spatiotemporal clusters of risk transmission of COVID-19 in Brazil. Their results showed higher COVID-19 risk in the northeastern metropolitan areas, as compared to the more rural parts of Brazil. An ecological study by Natividade et al. [43] investigated the effect of living conditions on social distancing in the COVID-19 pandemic in Salvador Bahia, Brazil, using local and global Moran’s I in QGIS, GeoDa, and R. Their results revealed that better living conditions were associated with a higher social distance index, as compared to areas with poor living conditions. Furthermore, de Souza et al. [39] used bivariate spatial correlation and multivariable spatial regression models in GeoDa to investigate the relationship of various indicators of human development and social vulnerability with COVID-19 incidence, mortality, and fatality rates in Brazil. They deduced that municipalities with low living conditions were highly exposed and would therefore need urgent attention to control the spread of disease.

### 3.3. Spatial Epidemiological Modeling

A rigorous spatial-statistical study was done by Cordes and Castro [48]. They used Moran’s I and scan statistics to identify clusters of COVID-19 testing rates and COVID-19 positivity rates in order to measure the urban health inequalities in New York, USA. They found negative associations of white race, education, and income with proportion of positive tests and positive associations with black race, Hispanic ethnicity, and poverty. Scarpone et al. [54] used many statistical and spatial techniques, including spatial autocorrelation, hotspot analysis in ArcGIS, and the R package spatstat, to demonstrate spatial associations of community interconnectedness, geographic location, transport infrastructure, and labor-market structure with COVID-19 incidence at the county scale in Germany. A study by Cuadros et al. [49] performed spatially explicit mathematical modeling of healthcare capacity and COVID-19. They argued that the higher COVID-19 attack rates in specific highly connected and urbanized regions could have significant implications for critical healthcare in these regions, notwithstanding their potentially high healthcare capacity compared to more rural and less connected areas in the US. Mizumoto et al. [52] used multivariable regression models to show that population density was statistically associated with COVID-19 cumulative morbidity rates and time delayed-adjusted case fatality rates in Italy and presented the geographic variability of these outcomes in choropleth maps.

Azevedo et al. [47] used geostatistical modeling to calculate spatial uncertainty of COVID-19 infection risk in Portugal. Ye and Hu [56] demonstrated the effectiveness of control measures of COVID-19 in the Yangtze River Delta region of China through spatial autocorrelation, polynomial regression, and hotspot analysis in ArcGIS. Another study by Mollalo et al. [45] used artificial neural networks in ArcGIS to show that ischemic heart disease, pancreatic cancer, and leukemia, along with household income and precipitation, were significant factors for predicting COVID-19 incidence rates in the USA. Likewise, another study by Maciel et al. [51] found positive associations between municipal human development index and COVID-19 in Brazil. They used spatial autocorrelation techniques in TerraView and GeoDa. Zhang and Schwartz [57] used ArcGIS to analyze the spatial disparities of COVID-19 in relation with socio-economic variables of urban and rural counties in the USA through multiple regression analysis. They found positive associations of population density, older age, and poverty with COVID-19 incidence and mortality. Xiong et al. [55] performed a detailed spatiotemporal investigation of environmental factors (land area, minimum elevation, maximum elevation, and range of elevation) and socio-economic factors (registered population, resident population, regional gross domestic production, and total retail sales of consumer goods) and COVID-19 at both prefecture and county levels in Hubei Province, China during the early months of epidemic. Spatial autocorrelation and Spearman’s rank correlation, executed in ArcGIS, were the main spatial statistical methods employed in this study. A study using inverse distance weighted (IDW) interpolation techniques and Pearson’s correlation in ArcGIS found that population density and asthma in urban areas and poverty and unemployment in rural areas were determinants of high COVID-19 mortality in Colorado, USA [53]. Similarly, Mollalo et al. [46] carried out multiscale geographically weighted regression modeling of COVID-19 incidence in the USA, in relation with socio-economic, demographic, behavioral, topographical and environmental factors. They found that income inequality was an influential factor in explaining COVID-19 incidence particularly in the tri-state area. The main spatial analysis software used was ArcGIS. Kraemer et al. [50] modeled population mobility in Wuhan and other provinces in China with R and found significant decrease in COVID-19 infections after implementation of governmental control measures to contain the disease.

## 4. Discussion

We reviewed and synthesized the use of spatial analysis tools and techniques in the context of the COVID-19 pandemic. Our review identified the application of spatial techniques, from simple disease mapping to the exploration of vulnerability factors and spatial epidemiological modeling. These studies not only tracked the changes in and intensity of COVID-19 spatial patterns, but also analyzed relationships with various potential influencing factors, such as socio-economic (occupation, income, transportation, population mobility, household density, government response, etc.), demographic (age, sex, ethnicity, nationality, etc.), environmental (temperature, topographic and built environment, etc.), and epidemiological and healthcare-related (tuberculosis incidence, social distancing, testing facilities, availability of ICU beds, health inequalities, etc.) variables. The findings of these studies are very important to our understanding of the spatial nature of the COVID-19 pandemic and enable the formulation of control strategies and allocation of appropriate healthcare measures to contain the disease.

Several studies used simple visualization techniques to present the spatial distribution of COVID-19, either by choropleth maps or dot density maps. Azevedo et al. [47] advocated the use of spatial statistics for the analysis of public health data, in comparison with more cartographic visualization, as spatial statistics were an improved way to measure disease risk and to reduce the bias of visual perception. Moreover, the authors claimed that spatial statistical techniques may facilitate the monitoring of governmental risk measures to contain the disease and to evaluate their efficiency [47].

Most of the spatial epidemiological studies were conducted during the initial months of the epidemic in China, USA, and Brazil. COVID-19 mapping with various other demographic, socio-economic, and environmental factors revealed and verified some basic facts about this new disease. For example, elderly male [31,32] and frontline health workers [24] showed high rates of infection and mortality. There was a direct relationship between living condition and maintaining social (i.e., physical) distancing [43]. Disease mapping revealed high transmission associated with high population mobility through road, rail, and air transport [35], suggesting that measures to reduce the mobility of people may be effective at controlling COVID-19 [28,50]. However, in terms of environment, no relationship was found between temperature and COVID-19 [21].

Furthermore, spatial analysis of COVID-19 also exhibited within-country variation, driven by various influencing factors [39]. For example, people living in disadvantaged areas were at higher risk of COVID-19 as compared to those living in more affluent areas. Particularly people living in areas with low socio-economic status, with poor access to sanitation and hand washing facilities, and those who were marginalized were at higher risk [42]. However, highly developed, high-density cities were at high risk too, and equity in testing and accessing healthcare facilities was critical in those urban neighborhoods that have high incidence rates of COVID-19 [48]. These studies demonstrate the need to prioritize deprived areas, both urban and rural, for infection control and healthcare [43].

In addition, COVID-19 cluster and hot spot studies were particularly useful for COVID-19 surveillance by identifying the size, location, and comparative risks of the disease. For example, daily cluster detection could track the emergence of hotspots of COVID-19 in the USA, as shown on a live web application for daily surveillance [25]. Authors called for locally specific interventions, in accordance with locally specific needs, to increase the effectiveness of emergency plans [22]. However, the application of these techniques was highly dependent on the availability of geographically fine-scaled data on demographic, environmental, epidemiological, and socio-economic characteristics.

In terms of disease modeling, most of the researchers used data-driven models instead of theory-driven methods. In multivariable regression models, strong positive correlations were found for socio-economic factors including population density, proportions of elderly residents, poverty, and percent population tested with COVID-19 morbidity and mortality [57].

According to the latest research on COVID-19 (not part of our review), the greater the number of transmissions, the more likely it is that new strains emerge and establish themselves in susceptible populations [58,59]. For example, as this manuscript is being written, two especially transmissible viral variants have become locally prevalent and are now spreading internationally, one in the United Kingdom, variant B.1.1.7, which probably emerged in September 2020 [58], and one in South Africa, the highly virulent variant 501.V2 [58].There is a need for future studies to detect and explore the potential impact of emerging variants on diagnostics, treatments, and vaccines [60]. Spatial epidemiological approaches may help enable early assessments of local variants and of the effectiveness of COVID-19 vaccines, and they may further contribute to the exploration of the economic consequences of this pandemic, if data and the global research agenda are coordinated effectively and efficiently across disciplines and international institutions.

However, in our study, data quality was the main limitation of any spatial analysis and determined the use of spatial techniques and methods. We observed great variation in the selected articles for the source, acquisition, and use of COVID-19 data along with demographic, social, and environmental variables. Many authors reported that the interpretation of their findings was limited by ecological fallacy [61]. Some limitations that were pointed out by researchers included bias of data because of low testing rate and asymptomatic population [43], underreporting of COVID-19 cases and deaths [39], and lack of locational data [40]. Because of such limitations, most of the studies did not explore causal relationships among COVID-19 variables [48]. Furthermore, availability of more detailed geographic data would allow analysis at finer spatial levels [29].

Our study also had some limitations. In our search for spatial studies of COVID-19, we used only the PubMed database. Our retrieval time was nine months (January–September 2020), and articles published after this time were not considered in this review. Furthermore, we only considered articles published in English; therefore, we had to exclude one article on the basis of language, although it did fulfill other eligibility criteria.

## 5. Conclusions

Notwithstanding these limitations, this scoping review extends existing reviews of spatial studies of the COVID-19 pandemic. Our review, by reflecting the application of most recent spatial techniques in visualization, exploration, and modeling of COVID-19, also showcased the most up-to-date trends in the field of health geography and spatial epidemiology. Cluster analysis through global and local Moran’s I, hotspot analysis, interpolation, and space–time scan statistics were found to be the main spatial techniques used to analyze COVID-19 data. However, the use of all these techniques was determined by the availability of spatial and relevant attribute data. We conclude that disease mapping has been used by researchers not only to detect spatial and temporal distribution, clusters, and hotspots of COVID-19, but also to explore relationships of COVID-19 with other factors. Exposure mapping has revealed that poor living conditions, elderly population, limited access to health facilities, and high population density were key risk factors for COVID-19 infections. Spatial epidemiological modeling has been used to explore and confirm positive associations between above mentioned socio-demographic factors (e.g., population density, proportion of elderly residents, poverty) and COVID-19 morbidity and mortality. We maintain that there is a need for spatial studies in other geographic areas of the world, since current studies were mainly focused on Asia and the Americas. We also call for a wider availability of health data at a fine-scaled geographic resolution (to the extent allowed by data-privacy rights) in order to facilitate application of the most advanced spatial approaches.

Spatial methods in epidemiology help to elucidate the spatial distribution of population health and the locally specific causes, so that we know where to intervene to prevent disease and promote health. This is particularly important in the current pandemic, as global public health measures to contain the disease (hand washing, social distance, mask wearing) need to be combined with locally targeted health interventions. Health-geographical approaches will continue to play a crucial role within the current pandemic and also beyond this crisis.

## Figures and Tables

**Figure 1 ijerph-18-02336-f001:**
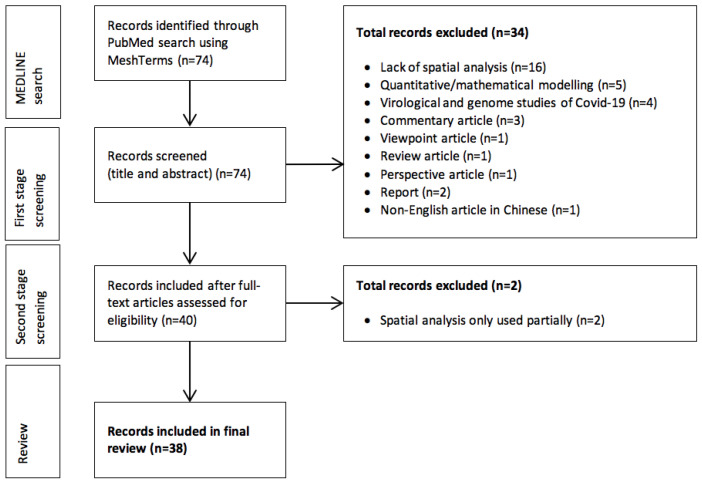
Flow chart showing the selection stages of the scoping review of spatial analysis used in COVID-19 studies (articles retrieved on 30 September 2020).

**Figure 2 ijerph-18-02336-f002:**
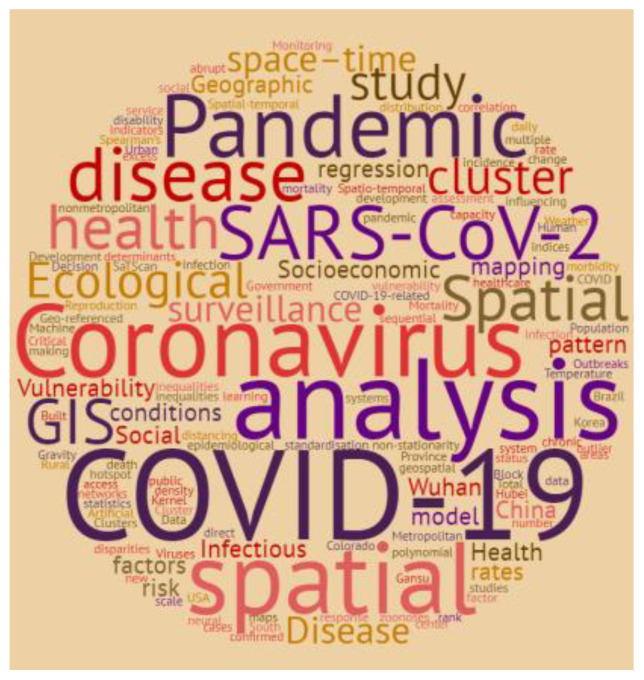
Cloud diagram showing the keywords of included articles.

**Table 1 ijerph-18-02336-t001:** Second-stage screening records and their study focus, main spatial techniques, software, and geographic extent, grouped by approach: disease mapping, exposure mapping, and spatial epidemiological modeling. SAC = spatial autocorrelation analysis.

#	Reference	Study Focus	Main Spatial Techniques	Main Software	GeographicExtent
**Disease Mapping**
1	Andrade et al. [20]	Space–time analysis of COVID-19	Prospective spatiotemporal scan statistic	QGIS 3.4.11,	Brazil
TerraView 4.2.2, SaTScan 9.6
2	Briz-Redón and Serrano-Aroca [21]	COVID-19 in relation with temperature	Choropleth maps of COVID-19 accumulated observed and predicted cases	R (automap),	Spain
Gstat

3	Cavalcante and Abreu [22]	Spatial distribution of COVID-19 cases and deaths	SAC with Moran’s I	QGIS 2.14.8,	Brazil
GeoDa 1.14.0
4	Fan et al. [23]	COVID-19 reported cases	Choropleth incidence map	ArcGIS 10.2.2	Gansu Province, China
LISA cluster analysis
5	Gao et al. [24]	COVID-19 in healthcare workers	Choropleth map showing distribution of COVID-19 infected health workers	QGIS 3.12	China
6	Hohl et al. [25]	COVID-19 space–time clusters through daily surveillance	Poisson space–time scan statistic	SaTScan,	USA
R, R Shiny
7	Huang et al. [26]	COVID-19 confirmed cases	Spatial autocorrelation (SAC) with Moran’s I	None mentioned	China
8	Jella et al. [27]	COVID-19 confirmed case with age and occupation	Overlay map of COVID-19 confirmed cases and orthopedic surgeons in patients >60 years of age	QGIS 3.12.1	USA
9	Jia et al. [28]	COVID-19 and population outflow	Overlay maps of population outflow from Wuhan and COVID-19 confirmed cases	ArcGIS 10.2	China
10	Kim & Castro [29]	Change in COVID-19 clusters according to government response	SAC with Moran’s I, Space–time scan statistic for spatio-temporal clusters of COVID-19	SaTScan 9.6,	South Korea
GeoDa 1.14,
ArcGIS 10.6.1
11	Li et al. [30]	Spatial analysis of COVID-19 clusters	SAC with Moran’s I	ArcGIS 10.4.1	China
12	Liao et al. [31]	COVID-19 cases (age, gender, nationality, occupation, and address)	Choropleth maps of COVID-19 confirmed case	ArcGIS 10.2	China
13	Michelozzi et al. [32]	COVID-19 mortality in relation with geographic area, age, and sex	Choropleth incidence map	R	Italy
14	Pedrosa and Albuquerque [33]	Spatial analysis of COVID-19 and healthcare services (number of ICU beds)	Case detection coefficient	Boxmap,	Brazil
SAC with Moran’s I (Bayesian method)	Moran Map
15	Ponjavić et al. [34]	Fast representation of COVID-19 through Geo visualization	Spatial visualization	ELIS (Epidemic Location Intelligence System)	Bosnia and Herzegovina
16	Rex et al. [35]	COVID-19 in relation with road and air transport	Kernel density estimation	QGIS 3.8	São Paulo, Brazil
17	Rivas et al. [36]	Spatial distribution of epidemic nodes and COVID-19 mortality	Spatial visualization of road rail and air connectivity	ArcGIS Pro 2.5.0,	China
SPSS, Minitab
18	Tang et al. [37]	Changing patterns of COVID-19	Choropleth maps of COVID-19 confirmed cases, SAC with Moran’s I	R,	China
ArcGIS 10.2
19	Yang et al. [38]	Spatiotemporal patterns of COVID-19	SAC with Moran’s I	ArcGIS 10.2	China
**Exposure Mapping**
20	de Souza et al. [39]	COVID-19 in relation with	Bivariate spatial correlation and multivariate and spatial regression models	GeoDa1.10.0.8	Brazil
living conditions
21	Gomes et al. [40]	Risk clusters of COVID-19 Transmission	SAC with Moran’s I	QGIS 3.4.11,	Brazil
SaTScan 9.6, TerraView 4.2.2
22	Lakhani [41]	Vulnerability assessment of COVID-19 by risk factors	Hotspot analysis (Getis-Ord Gi*)	ArcGIS 10.4.1	Australia
23	Macharia et al. [42]	Assessment of the vulnerability of COVID-19 (social and epidemiological)	Spatially overlaid via	Arc GIS 10.5,	Kenya
arithmetic mean and equally weighted	R 3.4.1
24	Natividade et al. [43]	Effect of living conditions on social distancing in COVID-19 pandemic	SAC with Moran’s I	QGIS 2.18,	Salvador-Bahia, Brazil
GeoDa 2.14,
R
25	Santos et al. [44]	COVID-19 vulnerability assessment (household density, old age population, tuberculosis incidence)	Choropleth maps of COVID-19 vulnerability Index	ArcGIS 10.5	Brazil
**Spatial Epidemiological Modeling**
26	A. Mollalo et al. [45]	Spatial and statistical prediction of COVID-19	Hotspot analysis by Getis–Ord Gi*	ArcGIS 10.4.1	USA
27	A. Mollalo et al. [46]	COVID-19 incidence relation with socio-economic demographic and environmental factors	Multiscale geographically weighted regression	ArcGIS 10.7	USA
28	Azevedo et al. [47]	New spatial methodology for spatial predictions assessment	Spatial Modeling,	ArcGIS online	Portugal
Stochastic simulations
29	Cordes and Castro [48]	COVID-19 and urban health inequalities	SAC with Moran’s I I	SaTScan9.6,	New York,
Pearson correlations	GeoDa 1.14.0,	USA
	ArcGIS 10.6.1,	
	R 3.6.2	
30	Cuadros et al. [49]	COVID-19 and healthcare capacity	Spatially-explicit mathematical modeling	ArcGIS 10.2	USA
31	Kraemer et al. [50]	Human mobility and control measures in relation with COVID-19	Generalized linear models (Poisson GLM, negative binomial GLM, log-linear regression)	R package,	China
GLMNET
32	Maciel et al. [51]	Spatial analysis of COVID-19 and its correlation with the municipal human development index (MHDI)	Bivariate LISA analysis,	TerraView 4.1.0,	Brazil
global Moran’s I	GeoDa
33	Mizumoto et al. [52]	COVID-19 confirmed cases	Choropleth maps of COVID-19 morbidity rates and crude fatality rates	R	Italy
COVID-19 crude case fatality ratio
34	Ramírez and Lee [53]	COVID-19 and social health determinants	Interpolation by inverse distance weighted (IDW), Pearson’s correlation	ArcGIS Pro	USA
35	Scarpone, et al. [54]	Spatial, socio-economic, and built-environment in relation to COVID-19 incidence	SAC with Moran’s I	ArcGIS 10.7.1,	Germany
R package spatstat
36	Xiong, et al. [55]	Spatial statistical analysis of COVID-19 and its Influencing factors	SAC with Moran’s I, Spearman’s rank correlation	ArcGIS 10.7	China
37	Ye and Hu [56]	Impacts of control measures on COVID-19 cases	Polynomial regression, SAC with Moran’s I	ArcGIS 10.4.1	China
38	Zhang and Schwartz [57]	Spatial pattern of COVID-19 in relation with socio-economic variables of urban and rural counties	Multiple regression analysis	ArcGIS 10.4.3	USA

**Table 2 ijerph-18-02336-t002:** Second-stage screening records and main study findings.

Reference	Main Findings
**Disease Mapping**	
Andrade et al. [20]	Active and emerging spatiotemporal clusters in southern central Sergipe, Brazil
Briz-Redón and Serrano-Aroca [21]	No evidence of a relationship between temperature and COVID-19 cases was found in Spain
Cavalcante and Abreu [22]	High risk of COVID-19 infection and deaths was found in neighborhoods in the South Zone of the city of Rio de Janeiro, Brazil
Fan et al. [23]	Spatial distribution of COVID-19 hotspots in China
Gao et al. [24]	Spatial distribution of COVID-19-infected healthcare workers in China, with Wuhan being the most severe, followed by Hubei Province and the rest of China
Hohl et al. [25]	As the pandemic progresses, the number of smaller clusters of remarkably steady relative risk increased in USA
Huang et al. [26]	Spatial patterns of COVID-19 in China, showing severe epidemic situation in Hubei province
Jella et al. [27]	Highest quintile of orthopaedic surgeons ≥60 years of age in New York, New Jersey, California, and Florida. These states were also most severely affected by COVID-19 in the USA
Jia et al. [28]	Spatial distribution of population outflow from Wuhan to the rest of China, evidence for an association between population outflow and COVID-19 cases
Kim & Castro [29]	South Korean government’s epidemic response measures were significantly associated with changes in COVID-19 clusters
Li et al. [30]	Provinces with high and low COVID-19 clusters in China, with Hubei as the only province with high-low aggregation
Liao et al. [31]	Strict preventive strategies aimed at the local culture, with inter-sectoral coordination and high degree of public cooperation, helped in controlling COVID-19 in Liangshan Prefecture, China
Michelozzi et al. [32]	Age and sex were confirmed as risk factors for COVID-19-related mortality in Italy, with elderly (aged 65+ years) and male persons exhibiting higher mortality
Pedrosa and Albuquerque [33]	Spatial distribution of intensive-care bed capacity was significantly associated with COVID-19 in Ceará, Brazil
Ponjavić et al. [34]	Distribution of COVID-19 incidence rates in Bosnia and Herzegovina
Rex et al. [35]	Metropoliton region of São Paulo State was a hotspot of COVID-19
Rivas et al. [36]	Network properties, including synchronicity and directionality, determined the epidemic profiles observed in several Chinese provinces, fostering the planning and implementation of more precise and locally specific interventions to control COVID-19
Tang et al. [37]	Spatial intensity of COVID-19 infection in China
Yang et al. [38]	Spatial clusters with high incidence rates were concentrated in Wuhan Metropolitan Area due to the high intensity of spatial interaction of the population
**Exposure Mapping**	
de Souza et al. [39]	Municipalities with low living conditions were highly exposed and would therefore need urgent attention to control the spread of disease in Brazil
Gomes et al. [40]	Higher COVID-19 risk in the northeastern metropolitan areas were found as compared to the more rural parts of Brazil
Lakhani [41]	Disability and access to health services were risk factors for COVID-19 in an elderly population in Melbourne, Australia
Macharia et al. [42]	COVID-19 risk was heterogeneously distributed across multiple social epidemiological indicators in Kenya
Natividade et al. [43]	Better living conditions were associated with a higher social distance index, as compared to areas with poor living conditions in Salvador Bahia, Brazil
Santos et al. [44]	City neighborhoods with higher average household density, high tuberculosis incidence, and large older populations (>60 years) were more vulnerable to COVID-19 infections in Rio De Janeiro, Brazil
**Spatial Epidemiological Modeling**	
A. Mollalo et al. [45]	Ischemic heart disease, pancreatic cancer, and leukemia, along with household income and precipitation were significant factors for predicting COVID-19 incidence rates in the USA
A. Mollalo et al. [46]	Income inequality was an influential factor in explaining COVID-19 incidence particularly in the tri-state area in the USA
Azevedo et al. [47]	Spatial uncertainty of COVID-19 infection risk was found in Portugal
Cordes and Castro [48]	Negative associations of white race, education, and income with proportion positive tests, and positive associations with black race, Hispanic ethnicity, and poverty in New York City, USA
Cuadros et al. [49]	Higher COVID-19 attack rates in specific highly connected and urbanised regions could have significant implications for critical healthcare in these regions, notwithstanding their potentially high healthcare capacity compared to more rural and less connected areas in the USA
Kraemer et al. [50]	Significant decrease in COVID-19 infections was found after implementation of governmental control measures to contain the disease in China
Maciel et al. [51]	There was a positive bivariate correlation between municipal human development index (MHDI) and the incidence of COVID-19 with the formation of a cluster in the metropolitan region of Fortaleza, Brazil
Mizumoto et al. [52]	Case fatality rates of COVID-19 estimates were statistically associated with population density and cumulative morbidity rate in Italy
Ramírez and Lee [53]	Population density and asthma in urban areas and poverty and unemployment in rural areas were determinants of high COVID-19 mortality in Colorado, USA
Scarpone et al. [54]	Location, densities of the built environment, and socio-economic variables were important predictors of COVID-19 incidence rates in Germany
Xiong et al. [55]	Social and economic development and population movement have strong impact on COVID-19 spread in Hubei province, China
Ye and Hu [56]	The effectiveness of control measures of COVID-19 in the Yangtze River Delta region of China
Zhang and Schwartz [57]	Positive associations were found among population density, older age, and poverty with COVID-19 incidence and mortality in urban and rural counties in the USA

## Data Availability

Not applicable.

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
