# Peer review of "Geospatial Analysis of COVID-19: A Scoping Review"

_ijerph, 2021, doi:10.3390/ijerph18052336_

Round 1

Reviewer 1 Report

 -In point 3 it would be interesting to add the results obtained in each of the investigations. This would make the article more complete. 
-The last paragraph of the discussion should be included in the conclusions. 
-Rewrite the conclusions based on the above. 

Author Response

Response to Reviewer #1

# 1: In point 3 it would be interesting to add the results obtained in each of the investigations. This would make the article more complete. 

Response 1: Thank you for this suggestion. We already had reported the results obtained in each investigation in our results section. However, we agree that it has added value to provide this information in a more focused form and therefore have added another table (table 2) for synthesizing the main results according to study area (disease mapping, exposure mapping, and spatial epidemiological modelling).

# 2: The last paragraph of the discussion should be included in the conclusions.

Response 2: Thank you for this comment. Following the author guidelines of MDPI, Int. Journal of Environmental Research and Public Health, we believe that our paragraph on the limitations (last paragraph of the discussion) is best placed in the discussion section. Therefore, we have not changed this last paragraph.

# 3: Rewrite the conclusions based on the above.

Response 3: Thank you, we have now included the main topical findings (reported in table 2) also in the conclusions.

Reviewer 2 Report

This article provides a synthesis of the techniques and associated findings in relation to Covid-19 and its geographic, environmental, and socio-demographic characteristics, following the PRISMA-ScR methodology for scoping reviews. It is important to reveal the nature of Covid-19 and its policies.

First, as we know,  avariant of the virus named UKB.1.1.7,  related with Covid-19, which started on December 1, 2020, is currently raging in the UK.  As a research review article, it  should be  no doubt to point out research frontier issues.  

Second,according to current research papers about Covid-19, the greater the number of transmissions, the more likely it is that a new strain will evolve, and it will promote selective infection and establish this strain in susceptible populations. The greater the number of transmissions, the more likely it is that new strains will evolve, and it will promote selective infection and establish this strain in susceptible populations. When we adopt the geospacial methods, how to reveal its internal mechanism ? At the economic level, how to deal with it?

Third, at present, the COVID-19 vaccine has been vaccinated, but the existing and other variant strains that have not yet emerged pose a threat to the effectiveness of the vaccine. For example, another highly toxic SARS-CoV-2 variant (South Africa V501) was found in South Africa. Like B.1.1.7, it spreads faster than other strains. This strain has quickly become a major infectious pathogen, and, like the group of Brazilian variants (B1.1.28) currently dominating the Amazon state, its mutations also occur in several regions of the viral spike protein. What is worrying is that, in the long run, mutations caused by mutations may cause strains to become resistant. 

In your revised manuscript, some information should be offered.

Thanks for reviewing your article.

Author Response

Response to Reviewer 2

# 1:       This article provides a synthesis of the techniques and associated findings in relation to Covid-19 and its geographic, environmental, and socio-demographic characteristics, following the PRISMA-ScR methodology for scoping reviews. It is important to reveal the nature of Covid-19 and its policies. First, as we know, a variant of the virus named UKB.1.1.7, related with Covid-19, which started on December 1, 2020, is currently raging in the UK. As a research review article, it should be no doubt to point out research frontier issues.

Response 1: Thank you. We have now added the related information with reference in the discussion section. Please see lines 319-330.

# 2:       Second, according to current research papers about Covid-19, the greater the number of transmissions, the more likely it is that a new strain will evolve, and it will promote selective infection and establish this strain in susceptible populations. When we adopt the geospacial methods, how to reveal its internal mechanism?

Response 2: Thank you. While our aim of this review was to report geospatial methods applied to the pandemic so far, we can only give some suggestions for future studies here. Please also see the newly added lines in the discussion section 319-330.

# 3:       At the economic level, how to deal with it.

Response 3: Thank you. Same as above, please see lines 319-330.

# 4:       Third, at present, the COVID-19 vaccine has been vaccinated, but the existing and other variant strains that have not yet emerged pose a threat to the effectiveness of the vaccine. For example, another highly toxic SARS-CoV-2 variant (South Africa V501) was found in South Africa. Like B.1.1.7, it spreads faster than other strains. This strain has quickly become a major infectious pathogen, and, like the group of Brazilian variants (B1.1.28) currently dominating the Amazon state, its mutations also occur in several regions of the viral spike protein. What is worrying is that, in the long run, mutations caused by mutations may cause strains to become resistant. In your revised manuscript, some information should be offered.

Response 4: Thank you also for this important comment. Please also see lines 319-330.

Reviewer 3 Report

The author(s) provides a synthesis of the techniques and associated findings related to Corvid-19 and its geographic, environmental, and socio-demographic characteristics, following the PRISMA-ScR methodology for scoping reviews. The research identifies current trends and research gaps. One of the main objectives was to review the use of spatial analysis tools used in investigations of the geographic variation of Covid-19. Second, to synthesize study results. Using a MEDLINE search via PubMed database, the authors identify 74 articles that could match their search criteria. From this sample, they were able to narrow down the search to 38 articles.

In some cases, data quality presented a problem to the authors. The search was limited to the PubMed database; even so, the paper provides a valuable contribution to the literature. It is well written, with only minor grammatical errors – mainly missing commas. The references are appropriate, although in some cases full details are not given – for example, Ref 4 should be 22 (1), 136-139. Ref 24 should be 15 (5): e0233255. Other references do not quote the issue number, and some references include the DOI number while others do not.  Fig. 1 depicts a well-defined research path. The word cloud in Fig.2 is visibly presented, and the 38 articles are clearly exhibited and analysed in the table. A good and timely paper.

Author Response

Response to Reviewer 3

# 1:       The author(s) provides a synthesis of the techniques and associated findings related to Corvid-19 and its geographic, environmental, and socio-demographic characteristics, following the PRISMA-ScR methodology for scoping reviews. The research identifies current trends and research gaps. One of the main objectives was to review the use of spatial analysis tools used in investigations of the geographic variation of Covid-19. Second, to synthesize study results. Using a MEDLINE search via PubMed database, the authors identify 74 articles that could match their search criteria. From this sample, they were able to narrow down the search to 38 articles. In some cases, data quality presented a problem to the authors. The search was limited to the PubMed database; even so, the paper provides a valuable contribution to the literature.

Response 1: Thank you for your comments.

# 2:       It is well written, with only minor grammatical errors – mainly missing commas.

Response 2: We have now corrected for the missing commas, and the grammar throughout the manuscript has been checked again by a native English speaker.

# 3:       The references are appropriate, although in some cases full details are not given – for example, Ref 4 should be 22 (1), 136-139. Ref 24 should be 15 (5): e0233255. Other references do not quote the issue number, and some references include the DOI number while others do not.

Response 3: Thank you for pointing out these errors. We have now corrected all references accordingly.

# 4:       Fig. 1 depicts a well-defined research path. The word cloud in Fig.2 is visibly presented, and the 38 articles are clearly exhibited and analysed in the table. A good and timely paper.

Response 4: Thank you for your valuable review!

Round 2

Reviewer 2 Report

In this revised manuscript, the authors make  their response to the reviewer.  And the revised manuscript contributes to the literature about  COVID-19. 

I think it should be accepted in present form.

Thank you for reviewing your manuscript.